# Increased Myocardial *MARK4* Expression in Patients with Heart Failure and Sleep-Disordered Breathing

**DOI:** 10.3390/ijms26083614

**Published:** 2025-04-11

**Authors:** Bettina Seydel, Philipp Hegner, Anna-Maria Lauerer, Sönke Schildt, Fatma Bayram, Maria Tafelmeier, Dominik Wermers, Leopold Rupprecht, Christof Schmid, Stefan Wagner, Lars Siegfried Maier, Michael Arzt, Simon Lebek

**Affiliations:** 1Department of Internal Medicine II, University Hospital Regensburg, 93053 Regensburg, Germany; bettina.seydel@ukr.de (B.S.);; 2Department of Cardiothoracic Surgery, University Hospital Regensburg, 93053 Regensburg, Germany

**Keywords:** MARK4, cardiovascular disease, heart failure, hypoxia, drug targets, biomarkers

## Abstract

Cardiovascular diseases are the leading cause of morbidity and mortality worldwide, underscoring the urgent need for novel therapeutic targets and strategies. The kinase MARK4 (MAP (microtubule-associated proteins)/microtubule affinity-regulating kinase 4) regulates microtubule-associated proteins pivotal for cell polarity, protein stability, and intracellular signaling. Animal models of heart failure revealed elevated MARK4 levels, which correlated with impaired cardiac contractility. However, the involvement of MARK4 and its potential as a molecular drug target has not yet been explored in the myocardium of cardiovascular patients. We investigated the *MARK4* mRNA expression in human myocardial biopsies of 152 high-risk cardiovascular patients undergoing cardiac surgery. Comprehensive echocardiography as well as testing for sleep-disordered breathing (SDB), a critical comorbidity in heart failure, were assessed preoperatively. We observed a substantial upregulation of myocardial *MARK4* expression in patients with impaired cardiac contractility, resulting in an inverse correlation with the left ventricular ejection fraction. Myocardial *MARK4* expression also correlated with echocardiographic E/e’, a central parameter of diastolic dysfunction. Mechanistically, our analyses revealed that *MARK4* expression increases in SDB and under hypoxic conditions, as evidenced by significant correlations between myocardial *MARK4* expression and factors like mean oxygen saturation, time with oxygen saturation below 90%, and the oxygen desaturation index. Multivariable regression analysis revealed that both left ventricular ejection fraction and mean oxygen saturation were independently associated with dysregulated *MARK4* levels, even when controlling for important clinical covariables as potential confounders. Taken together, our findings demonstrate that *MARK4* expression is highly increased in the myocardium of cardiovascular high-risk patients, suggesting it is a potential molecular target against cardiovascular diseases.

## 1. Introduction

Heart failure is a leading cause of morbidity and mortality worldwide [1]. In 2019, approximately 56.2 million individuals were diagnosed with heart failure [2]. The incidence of heart failure has been increasing (from 27.2 million in 1990), with a higher prevalence in older adults compared to the younger populations [2]. The most common etiologies of heart failure are ischemic heart disease followed by hypertensive heart disease [1,2]. Even though the hospitalization rate in older adults (and therefore the overall hospitalization rate) has been decreasing since the 1990s, it still remains high [3]. In contrast, the hospitalization rate continues to increase in younger people [3]. Despite advancements in medical treatments in recent years, therapeutic options remain limited [4,5] considering that the 5-year survival rate remains relatively low at 56.7% [6].

A highly prevalent comorbidity in cardiovascular high-risk patients is sleep-disordered breathing (SDB), which is associated with increased morbidity and mortality [7,8]. Despite its clinical significance, treatment options remain limited [9]. The standard therapy for obstructive sleep apnea, continuous positive airway pressure (CPAP), has been shown to improve ejection fraction in patients with heart failure but does not reduce hospitalization or mortality rates [10,11,12]. Therefore, the development of new, specific, and advanced therapies for both heart failure and SDB is crucial, which requires the identification of novel molecular targets.

MARK4 is a member of the MARK family (MAP/microtubule affinity-regulating kinase), which consists of 4 paralogs: MARK1, MARK2, MARK3, and MARK4 [13,14,15]. These kinases belong to the serine/threonine protein kinase family and play a crucial role in regulating microtubule-associated proteins (MAPs) through phosphorylation [15]. They are involved in the dynamics of microtubules and therefore influence cell polarity, protein stability, intracellular signaling, and the cell cycle [13,15]. MARK proteins are expressed in multiple organs [13,14,15] and are implicated in various diseases, including neurodegenerative disorders and cancer [16,17,18]. Recently, MARK4 has also been implicated in heart failure in mice [19,20]. In diabetic and ischemic cardiomyopathy, the expression of *Mark4* mRNA as well as the expression of MARK4 protein were significantly elevated [19,20]. Notably, the knockout of *Mark4* significantly alleviated the reduction in ejection fraction in ischemic heart failure [20].

Given these findings, MARK4 appears to be a promising molecular target for heart failure therapy. However, the regulation and expression of MARK4 in the human myocardium have not been described yet. Investigating *MARK4* in the human myocardium is crucial for validating its implication in cardiac function and disease in human patients, as emerging evidence suggests that this kinase may be involved in the regulation of myocardial function and stress responses. Thus, the aim of this study is to quantify the myocardial *MARK4* expression in 152 high-risk cardiovascular patients.

## 2. Results

### 2.1. Baseline Characteristics

A total of 948 patients, aged 18 to 85, who had given their written informed consent and underwent an elective coronary bypass operation were initially included. Patients were excluded due to withdrawal of consent (n = 16), insufficient polygraphy data (n = 117), lack of atrial appendage samples for PCR analysis (n = 632), or cancellation of the surgery (n = 31). Finally, a study cohort of 152 patients remained and was analyzed for myocardial *MARK4* mRNA expression (Figure 1).

Participants were stratified into two groups based on the median expression of *MARK4*: those with a higher expression (>median) and those with lower expression (<median). Overall, the study cohort represents a typical high-risk cardiovascular population (Table 1) [21]. The average age was 65.7 ± 9.0 years, and the majority of the patients were male (87.5%), which was comparable in both *MARK4* groups. Also, there was no significant difference in *MARK4* expression between male and female patients (in % of *ACTB*: 1.6 ± 1.0 vs. 1.3 ± 0.6, *p* = 0.216). In addition, patients of both *MARK4* groups frequently displayed several cardiovascular risk factors including obesity (body mass index [BMI]: 28.6 ± 4.5 kg/m^2^), diabetes mellitus, hyperlipidemia, and arterial hypertension. Patients with a higher *MARK4* expression were more frequent former smokers (68.7% vs. 50.0%, *p* = 0.028). The kidney function (i.e., GFR) was similar in both groups.

### 2.2. Myocardial MARK4 Expression Is Increased in Patients with Heart Failure

We observed a higher New York Heart Association (NYHA) status in patients with an increased *MARK4* expression (Table 1). The difference between NYHA III and IV was not statistically significant, likely due to the small sample size. The clinical symptoms are reflected in the results of the preoperative echocardiography, which were available for 123 patients of the study cohort (Table 2 and Figure 2).

We found a significantly decreased mean left ventricular ejection fraction (LVEF) in patients with a high myocardial *MARK4* expression (51.9 ± 12.3% vs. 57.4 ± 7.9%, *p* = 0.004). Correspondingly, the proportion of patients with a reduced LVEF (<55%) was more than doubled in patients with a high *MARK4* expression (Table 2). A comparison of the absolute number of patients with normal ejection fraction vs. reduced ejection fraction plotted according to the *MARK4* expression showed a rightward shift to higher *MARK4* values (Figure 2A). Accordingly, we detected significantly higher *MARK4* levels in patient groups with a decreased ejection fraction (Figure 2B). Linear regression analysis revealed a highly significant negative correlation between the LVEF and the *MARK4* expression (β = −0.044, 95% confidence interval: −0.059 to −0.030, *p* < 0.001) (Figure 2C). There was also a trend toward an increased mean left ventricular end-diastolic diameter (LVEDD) in patients with a higher *MARK4* expression (53.3 ± 7.2 mm vs. 50.8 ± 6.9 mm, *p* = 0.103), and linear regression analysis demonstrated a significant positive association between the LVEDD and *MARK4* expression (β = 0.036, 95% confidence interval: 0.005 to 0.068, *p* = 0.025) (Figure 2D). Additionally, the mean E/e’ ratio, a marker of diastolic dysfunction, was significantly elevated in patients with a higher *MARK4* expression (12.8 ± 5.5 vs. 9.3 ± 3.6, *p* = 0.005). Linear regression analysis revealed a significant positive correlation between the E/e’ ratio and *MARK4* expression (β = 0.067, 95% confidence interval: 0.003 to 0.131, *p* = 0.040) (Figure 2E). Furthermore, patients with a higher *MARK4* expression exhibited a significantly increased systolic pulmonary artery pressure (sPAP; 29.6 ± 10.6 mmHg vs. 23.8 ± 12.4 mmHg, *p* = 0.035). In line with these findings, we observed a strong trend (*p* = 0.070) toward increased N-terminal pro-B-type natriuretic peptide (NT-pro-BNP) levels, which were more than doubled in patients with a higher *MARK4* expression (Table 1). Plus, patients with a higher *MARK4* expression were treated with loop diuretics twice as often as those with a lower expression (*p* = 0.031, Table 1). We also detected slightly increased systemic inflammation levels in patients with a higher *MARK4* expression (i.e., elevated C-reactive protein), which was previously linked to systolic and diastolic cardiac dysfunction [22,23].

### 2.3. Myocardial MARK4 Expression Is Increased in Patients with SDB

All 152 patients of this study cohort underwent preoperative polygraphy to test for SDB (Table 3 and Figure 3), a highly prevalent disorder in cardiovascular high-risk patients [7,8]. In total, 73 out of the 152 patients (48%) had an apnea-hypopnea index (AHI) ≥ 15/h and were diagnosed with SDB. The analyses were conducted including all patients, both with and without SDB.

The total recording time was similar in both patients’ cohorts (Table 3). The prevalence of SDB was 1.5-fold increased in patients with a higher *MARK4* expression (57.9% vs. 38.2%, *p* = 0.020). In line with this, the mean AHI was significantly greater in the higher *MARK4* group (22.4 ± 17.8 vs. 14.0 ± 12.2, *p* = 0.002). While the obstructive apnea index and central apnea index were similar between both *MARK4* groups, we found evidence that intermittent hypoxia is critical in patients with an increased *MARK4* expression. We first analyzed whether there is an association between hypoxia parameters and the *MARK4* expression. Indeed, a comparison of the *MARK4* expression showed higher *MARK4* values in patients with an increased hypoxia burden, which was defined as mean peripheral oxygen saturation (SpO_2_) < 94% (Figure 3A,B).

Accordingly, the mean SpO_2_ was significantly lower in the higher *MARK4* expression group (91.8 ± 2.5% vs. 93.2 ± 1.8%, *p* < 0.001), as was the mean of minimum SpO_2_ (79.8 ± 7.2% vs. 82.7 ± 6.7%, *p* = 0.009) (Table 3). This resulted in a highly significant negative association between the mean SpO_2_ and the *MARK4* expression (β = −0.200, 95% confidence interval: −0.260 to −0.139, *p* < 0.001) (Figure 3C). Similarly, the mean nocturnal time with SpO_2_ < 90% was more than doubled in patients with a higher *MARK4* expression (17.8 ± 21.4 min vs. 6.9 ± 11.8 min, *p* < 0.001), and the mean oxygen desaturation index was increased by 1.7-fold (19.8 ± 17.1 vs. 11.4 ± 10.8, *p* < 0.001) (Table 3). Indeed, linear regression analyses confirmed significant positive correlations between the nocturnal time with SpO_2_ < 90% and *MARK4* expression (β = 0.028, 95% confidence interval: 0.021 to 0.036, *p* < 0.001), as well as between the oxygen desaturation index and *MARK4* expression (β = 0.024, 95% confidence interval: 0.014 to 0.033, *p* < 0.001) (Figure 3D,E). In summary, these observations indicate that chronic intermittent hypoxia is the main driver of *MARK4* upregulation in SDB.

### 2.4. Risk Factors for an Increased Myocardial MARK4 Expression

To evaluate the influence of several clinical covariates, we performed multiple logistic regression analyses with potential clinical confounders, including age, sex, obesity, renal function (i.e., GFR), and hemoglobin (Figure 4). None of these variables had a significant impact on the *MARK4* expression. Notably, patients with a LVEF < 55% had an >3.5-fold (*p* = 0.002) increased risk for an elevated *MARK4* expression (*p* = 0.002). Corresponding observations were made regarding an increased E/e’ as a parameter for diastolic dysfunction (odds ratio = 1.228, *p* = 0.025). Accordingly, treatment with loop diuretics was associated with an >2.5-fold increased risk for a high *MARK4* expression (*p* = 0.034). Moreover, the presence of SDB as well as a mean SpO_2_ < 94% were associated with an >2.2- (*p* = 0.016) and >2.5-fold (*p* = 0.014) increased risk for a higher *MARK4* expression, respectively. As shown in Figure 4, multiple clinical covariates appear to influence myocardial *MARK4* expression.

To address the effect of potential confounders, we conducted a comprehensive multivariable logistic regression analysis incorporating age, male sex, BMI, LVEF, E/e’, SDB, and the mean SpO_2_ (Table 4). Notably, only the LVEF and the mean SpO_2_ remained statistically significant in the multivariable model, indicating that both variables serve as independent predictors of myocardial *MARK4* expression.

## 3. Discussion

In this study, we investigated the expression of *MARK4* in human myocardial biopsies from 152 high-risk cardiovascular patients undergoing cardiac surgery. We demonstrated that impaired cardiac contractility (decreased LVEF) as well as nocturnal hypoxia (decreased mean SpO_2_) are independently associated with an increased myocardial *MARK4* expression, thereby suggesting *MARK4* as a potential molecular target for heart failure therapy.

Heart failure is defined as a clinical syndrome with specific symptoms (e.g., dyspnea, peripheral edema) resulting from structural and/or functional abnormalities of the heart [4,5]. Despite recent advancements in treatment, including the use of sodium-glucose co-transporter 2 inhibitors in addition to previously available angiotensin-converting enzyme inhibitors, angiotensin receptor blockers, angiotensin receptor–neprilysin inhibitors, mineralocorticoid receptor antagonists, beta-blockers, and diuretics to reduce congestion, as well as other adjunct therapies in select patients [4,5], heart failure remains a major cause of morbidity, mortality, and hospitalization worldwide [1,3,6]. Therefore, the development of new medical treatments with new molecular targets is urgently warranted.

The MARK family (MAP/microtubule affinity-regulating kinase) is a group of serine/threonine protein kinases consisting of four paralogs (MARK1, 2, 3, and 4) [13,14,15]. These proteins feature 5 domains: an N-terminal header, a catalytic domain for phosphorylation, a Ubiquitin-associated (UBA) domain that provides protein stability, a spacer region, and a C-terminal tail. The spacer domain is the most variable domain between the four paralogs [14,16,24,25]. MARKs are expressed in various organs like the brain, kidney, spleen, heart, skeletal muscle, and testis, and perform diverse functions. A primary role of these kinases is the regulation of microtubule dynamics, which influences cell polarity, protein stability, intracellular signaling, and the cell cycle [13,15]. These processes are regulated by phosphorylation of MAP2 and MAP4 through MARK, which leads to their detachment from microtubules thereby causing microtubule instability with disorganization of the microtubule array [15]. MARK dysregulation has been associated with multiple diseases. For example, MARK1 and MARK2 are implicated in the development of neurodegenerative diseases such as Alzheimer’s disease through phosphorylation of Tau, which leads to a detachment of Tau from microtubules and therefore to microtubule breakdown, disruption of the axonal transport, and Tau aggregation [16]. Additionally, MARK proteins participate in numerous signaling pathways and are associated with cancers. For instance, MARK2 is involved in the PI3K/AKT/NF-κB signaling pathway and its upregulation leads to resistance to cisplatin in osteosarcoma and lung cancer by decreasing apoptosis [17,18]. MARK3 is involved in the regulation of cell cycle progression and cellular proliferation through interaction with Cdc25C and 14-3-3 [26] and through phosphorylation of KSR1 in the KSR1/MEK complex and is therefore part of the Ras-pathway [27]. MARKs are also implicated in metabolic processes like glucose metabolism (MARK2) [28] and lipid metabolism (MARK3) [29].

MARK4, like other members of the MARK family, regulates MAPs through phosphorylation, thereby affecting microtubule dynamics [14,15]. Two isoforms, MARK4L and MARK4S, have been identified. MARK4L is the main isoform in most organs like the spleen, liver, lung, intestines, stomach, testis, and kidneys [30] while MARK4S is predominantly expressed in the brain. Both isoforms are equally expressed in the heart [14,30]. Notably, overexpression of both isoforms leads to a lower density of the microtubule network and even cell death [31]. Dysregulation of MARK4 has been associated with various diseases, including different types of cancer and Alzheimer’s disease. Plus, increased MARK4 expression was linked to increased oxidative stress as well as regulation of lipid and glucose metabolism [32,33,34,35,36,37]. Thus, the development of MARK4 inhibitors is the subject of current research given the potential as a therapeutic target in multiple diseases [38].

In cardiomyocytes, MARK4 phosphorylates MAP4, which enables tubulin carboxypeptidase vasohibin 2 (VASH 2) to access microtubules, which leads to cleavage of the C-terminal tyrosine residue of α-tubulin. Removal of the C-terminal tyrosine residue is commonly referred to as detyrosination [20,39,40]. Detyrosination results, among other effects, in increased microtubule stability by inhibiting their disassembly [41] and by forming a cross-linked network of microtubules and intermediate filaments through complexes with desmin [39]. In cardiomyocytes, microtubule detyrosination increases cellular stiffness and mechanical resistance to contraction, which results in impaired contractility [39]. In animal models of diabetic and ischemic cardiomyopathy, the expression of *Mark4* mRNA, as well as the expression of MARK4 protein and the levels of detyrosinated microtubules were significantly elevated [19,20]. Microtubule detyrosination was shown to inversely correlate with the left ventricular ejection fraction [20]. Knockdown of *Mark4* reduced MAP4 levels, which subsequently decreased microtubule detyrosination and alleviated the reduction in ejection fraction [19,20]. Additionally, the knockdown of *Mark4* in cardiomyocytes mitigated pathological changes, including oedema, increased intercellular space, cell fusion, fibrosis, as well as mitochondrial ultrastructural damage, and cardiomyocyte apoptosis [19]. Similarly in myocardial samples from patients with heart failure, MAP4 was significantly upregulated [41], and the levels of detyrosinated microtubules were markedly increased [39]. Consistent with the findings in animal models, the levels of detyrosinated microtubules in human myocardium inversely correlated with the left ventricular ejection fraction [39], and suppression of microtubule detyrosination reduced cardiomyocyte stiffness and improved cardiomyocyte contractility [41].

These findings suggest that MARK4 dysregulation is mechanistically involved in patients with heart failure. However, the expression of *MARK4* in human myocardium has not been investigated yet. In line with the above-mentioned data on MARK4 downstream targets, we observed higher *MARK4* expression levels in patients with a lower LVEF. Additionally, these patients exhibited higher E/e’ ratios, indicating diastolic dysfunction that could be attributed to myocardial “stiffness”.

Besides, in patients with a reduced LVEF, we also found a >2.2-fold increased risk for a high myocardial *MARK4* expression in patients with SDB. Notably, the prevalence of SDB is high in patients with heart failure and is associated with increased mortality [7,8]. The treatment options for both obstructive sleep apnea (OSA) and central sleep apnea (CSA) are also still limited [9]. The most common therapy for OSA is continuous positive airway pressure therapy (CPAP) [9]. In some cases, mandibular advancement devices, neurostimulation of the hypoglossus nerve, and surgical interventions can be considered [9]. For patients with CSA and a normal systolic left ventricular function (LVEF > 45%) CPAP therapy and adaptive servoventilation (ASV) are recommended. However, ASV was found to increase mortality in patients with reduced ejection fraction (LVEF ≤ 45%) and CSA [9,42], but not OSA [43]. CPAP therapy was shown to improve the ejection fraction in patients with heart failure and OSA or CSA [10,11,43] and enhances the quality of life in OSA patients, but it does not reduce hospitalization or mortality rates [11,12]. Other recent studies demonstrated that CPAP therapy failed to improve cardiovascular outcomes or reduce the burden of atrial fibrillation [44,45]. This could be attributed to limited treatment effectiveness as well as the frequently observed poor adherence to CPAP therapy among patients with cardiovascular disease [46]. These facts highlight the need for new therapeutic concepts in SDB, which may also include pharmacological approaches. We recently found an increased and pathogenic activity of the cardiac stress-responsive enzyme Ca^2+^/calmodulin-dependent protein kinase IIδ (CaMKIIδ) in the myocardium of mice and patients with SDB [47,48,49]. However, as targeting CaMKIIδ faces several limitations [50,51,52], we were eager to identify novel molecular targets involved in SDB.

In our study, we demonstrated that *MARK4* expression correlates with the level of hypoxia (e.g., nocturnal time spent below 90% SpO_2_). This is congruent with the results of previous studies in mice with ischaemic heart [20] and brain disease [14] that observed increased levels of MARK4 with subsequently reduced cell viability.

Previous animal studies demonstrated that MARK4 is increased in cardiac pathology and that targeting this kinase is a promising therapeutic approach. We can now validate that myocardial *MARK4* expression is significantly increased in patients with impaired systolic and diastolic cardiac function, which we found to be independent of other clinical parameters.

The development of MARK4 inhibitors is the subject of current research but primarily in the context of neurodegenerative diseases and cancer rather than heart failure. So far, no traditional compound-based inhibitors of MARK4 have been tested in cardiomyocytes. However, blocking MARK4 with traditional compound-based inhibitors would be challenging due to the high structural similarity among the various MARK paralogs. Considering their diverse functions, such as regulation of the cell cycle [27], axonal transport in neurons [53], or regulation of the glucose metabolism [29], as well as the widespread expression of MARK4 in various tissues including heart, brain, spleen, liver, lung and intestines [30], a paralog-unspecific or global MARK4 inhibition carries the risk of unforeseen side effects. The potential side effects could range from hypoketotic hypoglycemia due to hepatic glycogen depletion (MARK3) [29] to loss of cell-cell interactions and reduction in cell proliferation with unpredictable adverse effects in the above-mentioned tissues that express MARK4 [31]. Therefore, a targeted approach blocking specifically MARK4 and exclusively in cardiomyocytes will be essential to avoid severe side effects. Potential strategies include CRISPR-based approaches for gene modulation or microRNA-based therapies. For example, miR199a-5p has been shown to cause a decrease in MARK4 protein expression in cardiomyocytes [54]. Further research, including in vivo animal studies and patient-derived cardiomyocyte models, will be necessary to provide evidence for its therapeutic potential. However, even the development of a highly selective drug with exceptional paralog- and tissue-specificity would require heart failure patients to add yet another pill to their daily regimen. Unfortunately, each additional medication tends to lower patient adherence, which is a critical concern in cardiovascular care. This drop in compliance not only undermines therapeutic success but also imposes a significant financial strain [55,56,57]. Thus, advanced biotechnological approaches are needed to address the challenges of achieving high on-target and tissue specificity, coupled with sustained MARK4 inhibition [58].

As this study is a retrospective analysis, it entails several limitations. First, we stratified groups based on the median *MARK4* expression. The median was specifically selected as a cutoff because it is more robust against extreme values, which are commonly encountered in biological data such as gene expression levels. While this simplifies statistical comparisons, such an approach may introduce bias, especially when a variable does not follow a normal distribution, which is the case for *MARK4* and a limitation of this study. To address this concern, we have incorporated several additional measures. First, we also compared *MARK4* expression in patient groups stratified by the LVEF as well as stratified by the presence of SDB. Plus, we performed various uni- and multivariable regression analyses based on the continuous *MARK4* expression levels. Notably, all these analyses support our conclusion that *MARK4* expression is increased in patients with a lower LVEF and SDB. To obtain human myocardial biopsies, we analyzed patients undergoing coronary artery bypass grafting. This means that all patients had coronary artery disease and its associated comorbidities, which led to a homogenous and potentially biased study population. This is a limitation of our study and reduces the generalizability of our findings to the broader heart failure population as other etiologies (e.g., non-ischemic heart disease) are underrepresented. To further validate and gain deeper mechanistic insights into the relationship between MARK4, cardiac function, and hypoxia, future studies involving larger and more diverse patient cohorts, predefined clinical endpoints, and experiments using pharmacological MARK4 inhibitors are required.

In conclusion, our study demonstrates a significant upregulation of myocardial *MARK4* expression in a well-characterized cohort of 152 cardiovascular high-risk patients undergoing elective coronary artery bypass grafting. This elevation correlates strongly with both impaired cardiac contractility and SDB. Furthermore, the inverse relationship between *MARK4* expression and LVEF, alongside its induction under hypoxic conditions, suggests an important role in heart failure pathophysiology. Thus, MARK4 represents a promising molecular target for therapeutic inhibition. Given the high structural and functional similarity of MARK4 to other kinases (particularly within the MARK family), precise targeting strategies are essential to minimize off-target effects and mitigate the risks of systemic inhibition with subsequent adverse side effects. Advanced biotechnological tools that allow tissue- and target-specific modulation will therefore be indispensable for harnessing the therapeutic potential of MARK4. Future investigations should further elucidate the detailed molecular mechanisms governing MARK4 activity, its downstream signaling pathways, and its interplay with critical cardiac and respiratory factors as well as sex-related effects. Another open question is whether CPAP therapy or oxygen supplementation reduces MARK4 expression and improves cardiac function. Such efforts will facilitate the identification of novel drug candidates and therapeutic strategies aimed at more precise and effective management of heart failure and its comorbidities.

## 4. Materials and Methods

### 4.1. Study Design

This is a cross-sectional study with a retrospective analysis of patients enrolled in the “Impact of sleep-disordered breathing on atrial fibrillation and perioperative complications in patients undergoing coronary artery bypass grafting surgery - a prospective observational study” (CONSIDER-AF: NCT02877745) study [59,60]. The study received approval from the local ethics committee (University of Regensburg, Bavaria, Germany; 15-101-0238) and follows the Declaration of Helsinki (1964, most recent revision 2013). Written informed consent was obtained from each patient prior to their inclusion in the study. Patient information and data can be shared only after each patient has provided informed consent for a specific request.

All patients undergoing cardiac surgery at the University Hospital Regensburg were screened for eligibility between May 2016 and September 2023. Exclusion criteria were severe obstructive pulmonary diseases, oxygen therapy, nocturnal positive airway pressure support or mechanical ventilation, pre-existing treated SDB, and preoperative use of inotropes or an intra-aortic balloon pump [59,60].

The term “sex” refers to a constellation of biological characteristics associated with specific anatomical and physiological features, such as gene expression, hormone levels, and chromosomes. It is typically categorized as male and female and assigned at birth.

### 4.2. Assessment of SDB

Each patient underwent testing for SDB in the preoperative night using polygraphy (SOMNOtouch™ RESP; SOMNOmedics, Randersacker, Germany), as previously described [59,60]. The following parameters were monitored: nasal flow, pulse oximetry, and thoracic breathing effort (using inductance plethysmography). The apnea-hypopnea index (AHI) quantifies the occurrence of apneas and hypopneas per hour of sleep, where apneas are defined as a ≥90% decline in airflow for ≥10 s and hypopnea as a decline in airflow by 30–90% compared to baseline for ≥10 s. SDB was diagnosed by an AHI ≥ 15 events per hour of recording time. Desaturations were defined as a decrease in oxygen saturation of ≥4%, and the oxygen desaturation index (ODI) was calculated as the frequency of desaturations per hour of recording time [59,60].

### 4.3. RNA Isolation, Transcription into cDNA, and Quantification of MARK4

Right atrial appendage biopsies were obtained intraoperatively and placed into an ice-cold Custodiol^®^ solution containing 2 mmol/L butanedione monoxime. The samples were then frozen in liquid nitrogen and stored at −80 °C. The RNA was isolated using the RNeasy Mini Kit (Qiagen, Hilden, Germany, catalog number 74106). The transcription of 1 µg RNA into cDNA was carried out using random primers (Promega Corporation, Madison, WI, USA catalog number C1181), PCR nucleotide mix (Promega, catalog number C1145), RNasin^®^ ribonuclease inhibitor (Promega, catalog number N2115), reverse transcriptase (Promega, catalog number M170B), and reverse transcriptase 5x reaction buffer (Promega, catalog number M531A) for 1 h at 37 °C.

To quantify the *MARK4* expression, TaqMan™ Fast Advanced Master Mix (Applied Biosystems, Waltham, MA USA, catalog number 4444557) and pre-designed TaqMan^®^ Gene Expression Assays (Applied Biosystems, catalog number 4331182) for *MARK4* (assay ID: Hs00230039_m1) and *ACTB* (assay ID: Hs01060665_g1) were used. We used *ACTB* (gene encoding for β-actin), which we previously found to be a viable housekeeper gene in the myocardium of cardiovascular patients [22,61]. Real-time qPCR was performed using the synthesized cDNA on a ViiA 7 real-time PCR system (Applied Biosystems). The reaction included an initial incubation with uracil-N-glycosylase at 50 °C (2 min), followed by polymerase activation at 95 °C (2 min). Then, 40 cycles were run at 95 °C (1 s) and 60 °C (20 s). Each sample was analyzed in triplicate, and the mean threshold cycle (Ct) was utilized for the comparative Ct relative quantification method [62]. To calculate the delta-Ct (dCt) value, the mean Ct value of the housekeeping gene (*ACTB*) of each sample was subtracted from the corresponding mean Ct value of *MARK4*. The relative expression of *MARK4* mRNA was determined by applying the formula 2^−dCt^ × 100, with the results presented as a percentage of *ACTB*.

### 4.4. Statistical Analyses

Statistical analyses were performed with GraphPad Prism Version 10.4.2 (GraphPad Software, Boston, MA, USA) and SPSS Version 28.0.1.1 (IBM, Armonk, NY, USA). Data were classified as either continuous or categorical. Continuous data are presented as mean ± standard deviation, while categorical variables are presented as absolute values and relative frequencies. Patients were stratified into two groups based on the median value of *MARK4* expression. One group included patients with a *MARK4* expression above the median, while the other group consisted of patients with a *MARK4* expression below the median. The Shapiro-Wilk test was used to assess the normality of data distribution. To compare the two groups, an unpaired *t*-test was used for normally distributed data, and the Mann-Whitney-U test for non-normally distributed data. One-way ANOVA with Holm-Šídák’s post-hoc test was used to compare more than two groups. Categorial variables were evaluated using the Chi-square test or Fisher‘s exact test for variables with a small sample size. Correlations were tested by linear regression analyses. Uni- and multivariable logistic regression analyses were performed with *MARK4* expression as the dependent variable and important clinical covariates as independent variables to evaluate the impact of potential confounders on the myocardial *MARK4* expression. A two-sided *p*-value below 0.05 was considered statistically significant.

## Figures and Tables

**Figure 1 ijms-26-03614-f001:**
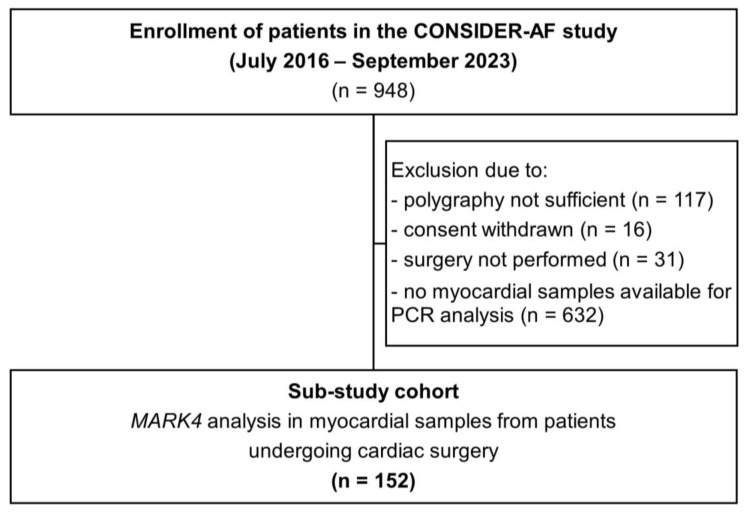
Flowchart showing patient enrollment.

**Figure 2 ijms-26-03614-f002:**
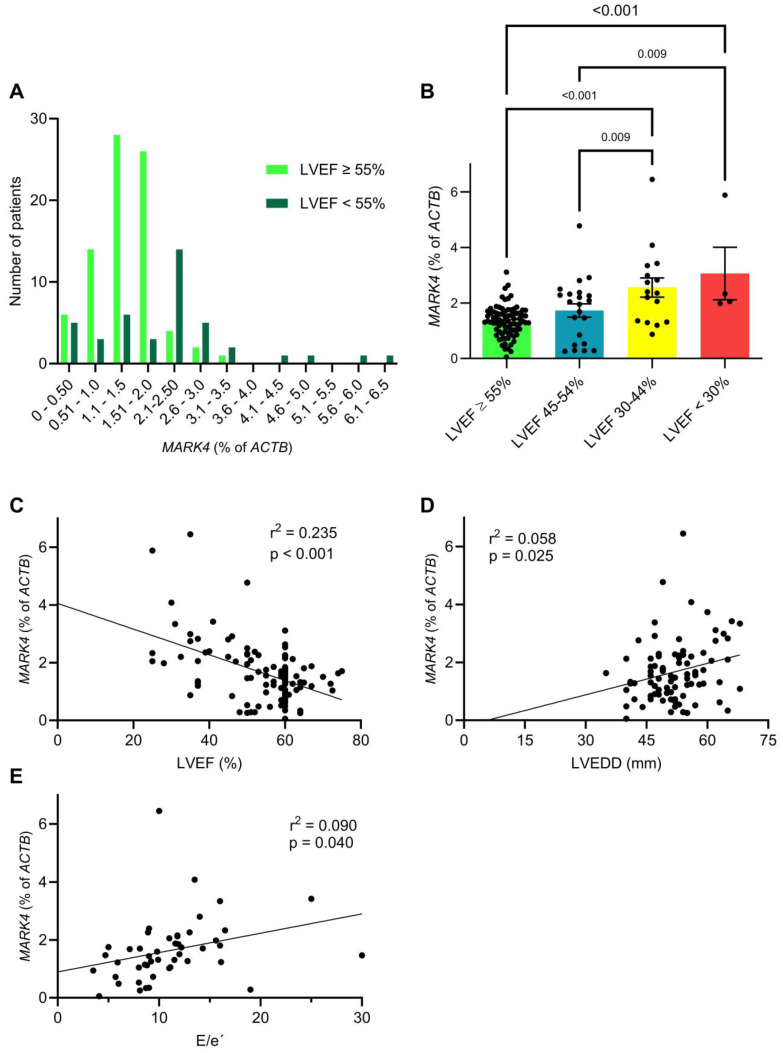
Myocardial *MARK4* expression is increased in patients with systolic and diastolic cardiac dysfunction. (**A**) Frequency distribution of patients with a normal (≥55%, n = 81) and a reduced (<55%, n = 42) left ventricular ejection fraction (LVEF), based on the *MARK4* expression level. (**B**) Mean *MARK4* levels in different LVEF categories (LVEF ≥ 55%, n = 81; LVEF 45–54%, n = 22; LVEF 30–44%, n = 16; LVEF < 30%, n = 4). (**C**) Linear regression analysis between the LVEF and the *MARK4* expression (n = 123). (**D**) Linear regression analysis between the left ventricular end-diastolic diameter (LVEDD) and the *MARK4* expression (n = 87). (**E**) Linear regression analysis between the E/e’ ratio and the *MARK4* expression (n = 47). Statistical comparisons are based on one-way ANOVA with Holm-Šídák’s post-hoc test (**B**) and simple linear regression analyses (**C**–**E**).

**Figure 3 ijms-26-03614-f003:**
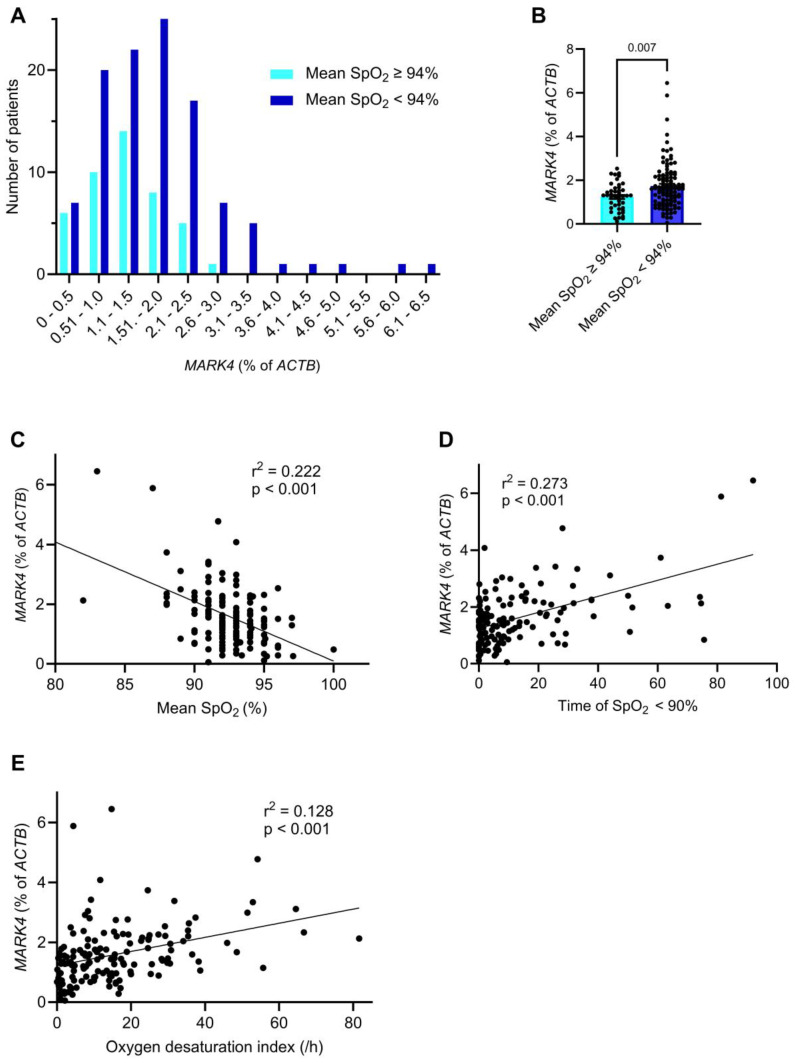
Myocardial *MARK4* expression is increased upon hypoxia. (**A**) Frequency distribution of patients with a normal (≥94%, n = 44) and a reduced (<94%, n = 108) mean oxygen saturation (SpO_2_), based on the *MARK4* expression level. (**B**) Mean *MARK4* levels in patients with a normal (≥94%, n = 44) and a reduced (<94%, n = 108) mean SpO_2_. (**C**) Linear regression analysis between the mean SpO_2_ and the *MARK4* expression (n = 152). (**D**) Linear regression analysis between the time of a SpO_2_ < 90% and the *MARK4* expression (n = 152). (**E**) Linear regression analysis between the oxygen desaturation index and the *MARK4* expression (n = 152). Statistical comparisons are based on a Mann-Whitney-U test (**B**) and simple linear regression analyses (**C**–**E**).

**Figure 4 ijms-26-03614-f004:**
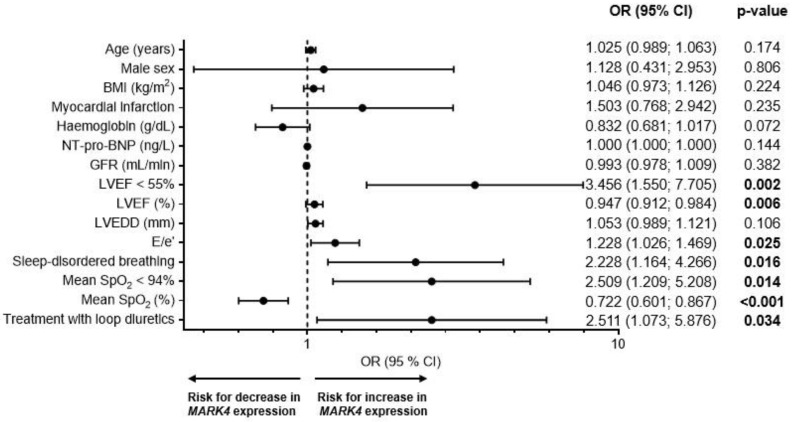
Risk factors for an increased myocardial *MARK4* expression. Forest plot showing the odds ratios (ORs) and 95% confidence intervals (CIs), based on the results of univariable logistic regression analyses of clinical covariates affecting the myocardial *MARK4* expression (n = 152).

**Table 1 ijms-26-03614-t001:** Baseline characteristics of the study population based on the *MARK4* expression.

	Total Cohort (n = 152)	<Median of *MARK4* (n = 76)	>Median of *MARK4* (n = 76)	*p*-Value
Age (years), mean ± SD	65.7 ± 9.0	64.7 ± 8.8	66.7 ± 9.2	0.119 ^MWU^
Male sex, n (%)	133 (87.5)	66 (86.8)	67 (88.2)	0.806 ^Chi^
BMI (kg/m^2^) mean ± SD	28.6 ± 4.5	28.1 ± 4.1	29.0 ± 4.8	0.224 ^T^
NYHA functional class, n (%)				
I	44 (28.9)	32 (42.1)	12 (15.8)	<0.001 ^Chi^
II	67 (44.1)	27 (35.5)	40 (52.6)	0.034 ^Chi^
III	39 (26)	17 (22)	22 (29)	0.353 ^Chi^
IV	2 (1)	0 (0)	2 (3)	0.492 ^F^
Cardiovascular risk factors				
Arterial hypertension, n (%)	127 (83.6)	65 (85.5)	62 (81.6)	0.512 ^Chi^
Systolic blood pressure (mmHg), mean ± SD	132.5 ± 17.4	134.5 ± 17.9	130.6 ± 16.2	0.144 ^MWU^
Diastolic blood pressure (mmHg), mean ± SD	77.2 ± 8.7	77.6 ± 8.1	76.9 ± 9.2	0.552 ^MWU^
Smoker, n (%)	98 (64.5)	43 (56.5)	55 (72.4)	0.042 ^Chi^
Previous smoker	79 (59.4)	33 (50.0)	46 (68.7)	0.028 ^Chi^
Current smoker	18 (25.0)	10 (23.3)	8 (27.6)	0.677 ^Chi^
Diabetes mellitus, n (%)	55 (36.2)	26 (34.2)	29 (38.2)	0.613 ^Chi^
HbA1c (%), mean ± SD	6.4 ± 1.3	6.4 ± 1.4	6.4 ± 1.2	0.505 ^MWU^
Hyperlipidaemia, n (%)	105 (69.5)	51 (67.1)	54 (72.0)	0.513 ^Chi^
History of TIA or Stroke, n (%)	25 (16.4)	15 (19.7)	10 (13.2)	0.274 ^Chi^
Atrial fibrillation, n (%)	20 (13.6)	9 (12.3)	11 (14.9)	0.654 ^Chi^
Postoperative atrial fibrillation, n (%)	33 (22.3)	12 (16.4)	21 (28.0)	0.091 ^Chi^
Haemoglobin (g/dL), mean ± SD	14.1 ± 1.6	14.3 ± 1.7	13.9 ± 1.6	0.070 ^T^
NT-pro-BNP (ng/L), mean ± SD	1165.7 ± 3233.9	726.4 ± 1307.7	1604.9 ± 4356.2	0.070 ^MWU^
GFR (mL/min), mean ± SD	75.0 ± 20.7	76.5 ± 20.1	73.5 ± 21.4	0.331 ^MWU^
CRP (mg/L), mean ± SD	6.9 ± 15.9	6.0 ± 13.0	7.8 ± 18.3	0.006 ^MWU^
Medical treatment, n (%)				
Angiotensin-converting enzyme inhibitors	70 (51.5)	38 (55.1)	32 (47.8)	0.394 ^Chi^
Angiotensin receptor blockers	35 (25.7)	16 (23.2)	19 (28.4)	0.491 ^Chi^
Calcium Channel Blocker	40 (29.4)	19 (27.5)	21 (31.3)	0.626 ^Chi^
Beta-blockers	90 (66.2)	43 (62.3)	47 (70.1)	0.957 ^Chi^
Mineralocorticoid receptor antagonists	12 (8.8)	6 (8.7)	6 (8.8)	0.335 ^Chi^
Loop diuretics	30 (22.1)	10 (14.5)	20 (29.9)	0.031 ^Chi^
Thiazide diuretics	26 (19.1)	10 (14.5)	16 (23.9)	0.164 ^Chi^

BMI, Body mass index; Chi, Chi-square test; CRP, C-reactive protein; F, Fisher’s exact test; GFR, Glomerular filtration rate; HbA1c, Glycated hemoglobin A1c; MWU, Mann-Whitney U test; NT-pro-BNP, N-terminal pro-B-type natriuretic peptide; NYHA, New York Heart Association status; SD, standard deviation; T, Student’s *t*-test; TIA, Transient ischemic attack.

**Table 2 ijms-26-03614-t002:** Comparison of echocardiographic assessment stratified by *MARK4* expression.

	Total Cohort (n = 123)	<Median of *MARK4* (n = 59)	>Median of *MARK4* (n = 64)	*p*-Value
LVEF (%), mean ± SD	54.5 ± 10.7	57.4 ± 7.9	51.9 ± 12.3	0.004 ^MWU^
LVEF < 55%, n (%)	42 (34.1)	12 (20.3)	30 (46.9)	0.002 ^Chi^
LVEDD (mm), mean ± SD	51.2 ± 9.2	50.8 ± 6.7	53.3 ± 7.2	0.103 ^T^
LAVI (mL/m^2^) mean ± SD	33.5 ± 12.9	31.5 ±11.2	35.7 ± 14.6	0.320 ^T^
E/e’ ratio, mean ± SD	11.1 ± 5.0	9.3 ± 3.6	12.8 ± 5.5	0.005 ^MWU^
sPAP (mmHg), mean ± SD	26.8 ± 11.8	23.8 ± 12.4	29.6 ± 10.6	0.035 ^MWU^
Vena cava inferior (mm), mean ± SD	15.3 ± 4.2	14.9 ± 4.2	15.9 ± 4.2	0.325 ^T^

Chi, Chi-square test; LAVI, Left atrial volume index; LVEDD, Left ventricular end-diastolic diameter; LVEF, Left ventricular ejection fraction; MWU, Mann-Whitney U test; sPAP, Systolic pulmonary artery pressure; T, Student’s *t*-test.

**Table 3 ijms-26-03614-t003:** Results of the preoperative polygraphy stratified by *MARK4* expression.

	Total Cohort (n = 152)	<Median of *MARK4* (n = 76)	>Median of *MARK4* (n = 76)	*p*-Value
Total recording time (min), mean ± SD	481.0 ± 51.8	481.4 ± 63.2	480.5 ± 37.5	0.137 ^MWU^
Sleep-disordered breathing, n (%)	73 (48.0)	29 (38.2)	44 (57.9)	0.015 ^Chi^
AHI (/h), mean ± SD	18.2 ± 15.8	14.0 ± 12.2	22.4 ± 17.8	0.002 ^MWU^
Obstructive apnea index (/h), mean ± SD	5.1 ± 8.7	3.9 ± 5.0	6.3 ± 11.2	0.123 ^MWU^
Central apnea index (/h), mean ± SD	6.0 ± 8.6	4.8 ± 7.1	7.2 ± 9.7	0.283 ^MWU^
Mean SpO_2_ (%), mean ± SD	92.5 ± 2.3	93.2 ± 1.8	91.8 ± 2.5	<0.001 ^MWU^
Minimum SpO_2_ (%), mean ± SD	81.2 ± 7.1	82.7 ± 6.7	79.8 ± 7.2	0.009 ^MWU^
Time of SpO_2_ < 90% (min), mean ± SD	12.3 ± 18.1	6.9 ± 11.8	17.8 ± 21.4	<0.001 ^MWU^
Oxygen desaturation index (/h), mean ± SD	15.6 ± 14.9	11.4 ± 10.8	19.8 ± 17.1	<0.001 ^MWU^
Mean heart rates (/min), mean ± SD	71.2 ± 14.3	69.3 ± 11.1	73.2 ± 16.8	0.139 ^MWU^

AHI, Apnea-hypopnea index; Chi, Chi-square test; MWU, Mann-Whitney U test.

**Table 4 ijms-26-03614-t004:** Multivariable logistic regression analysis with several clinical covariates.

Multivariable Model for an Increased *MARK4* Expression r² = 0.188, n = 121	OR (95% CI)	*p*-Value
Age (years)	1.014 (0.970; 1.061)	0.538
Male sex	0.543 (0.149; 1.978)	0.355
BMI (kg/m^2^)	1.008 (0.915; 1.111)	0.867
LVEF (%)	0.962 (0.927; 0.998)	0.039
E/e’	0.999 (0.992; 1.007)	0.868
Existing SDB	1.546 (0.661; 3.613)	0.315
Mean SpO_2_ (%)	0.798 (0.652; 0.976)	0.028

BMI, Body mass index; LVEF, Left ventricular ejection fraction; SDB, Sleep-disordered breathing; SpO_2_, Peripheral oxygen saturation.

## Data Availability

The original contributions presented in this study are included in the article. Further inquiries can be directed to the corresponding author.

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
