# Peer review of "Increased Myocardial MARK4 Expression in Patients with Heart Failure and Sleep-Disordered Breathing"

_ijms, 2025, doi:10.3390/ijms26083614_

Round 1
Reviewer 1 Report
Comments and Suggestions for Authors
The study "Increased Myocardial MARK4 Expression in Patients with Heart Failure and Sleep-Disordered Breathing" delves into the intriguing role of MARK4 in cardiovascular disease, particularly in patients experiencing both heart failure and sleep-disordered breathing (SDB). The authors present compelling evidence that heightened MARK4 expression in cardiac tissue correlates with impaired cardiac contractility and hypoxia, making it a potential therapeutic target. This research stands out because it bridges molecular insights with clinical observations, shedding light on a pathway that has not been extensively explored in human heart failure.
One of the study’s major strengths is its well-defined patient cohort. With 152 individuals at high cardiovascular risk, the findings hold relevance beyond a niche subset of patients, making the results more translatable to real-world clinical settings. The statistical analysis is rigorous, utilizing appropriate regression models to examine the relationship between MARK4 expression and key echocardiographic markers such as left ventricular ejection fraction (LVEF) and E/e’ ratio. By incorporating multivariable adjustments, the authors ensure that the associations observed are not merely artifacts of confounding variables like age, BMI, or renal function. The study also benefits from a multifaceted approach, analyzing both myocardial function and the impact of SDB-related hypoxia, offering a more comprehensive picture of how MARK4 fits into the pathophysiology of heart failure. Moreover, their methodology for RNA quantification is solid, using real-time quantitative PCR (qPCR) with ACTB as a housekeeping gene, which strengthens the reliability of the molecular findings.
That said, there are several areas where the study could be expanded to increase its impact. While the data strongly suggest a link between elevated MARK4 and worsened cardiac function, causality remains an open question. Future studies using functional experiments, such as siRNA knockdown of MARK4 in cardiomyocytes or organoid models, would be invaluable in dissecting this cause-effect relationship. Additionally, exploring how MARK4 interacts with other key cardiac proteins—particularly those involved in cytoskeletal integrity and contractile function—would provide a more mechanistic understanding of its role in heart failure.
Another aspect worth exploring is the potential influence of sex differences. The study cohort is overwhelmingly male (87.5%), yet cardiovascular disease often presents differently in men and women. Could MARK4 expression or its effects on cardiac function vary by sex? This is an underappreciated aspect of many cardiovascular studies and warrants further investigation.
Overall, this study makes a significant contribution to our understanding of MARK4 in cardiovascular disease, particularly in the intersection of heart failure and SDB. The statistical analysis is robust, and the findings have strong clinical relevance. However, further mechanistic studies, a deeper exploration of therapeutic strategies, and additional experimental validation would strengthen the case for targeting MARK4 in heart failure treatment. Despite these limitations, this research lays a solid foundation for future investigations and opens intriguing new avenues for therapeutic intervention.
Please answer the following questions by expanding the discussion.
Are there any existing inhibitors of MARK4 that have been tested in cardiomyocytes, and if not, what potential drug candidates could be explored?
Is increased MARK4 expression a consequence of heart failure, or does it contribute to its progression?
Could genetic or pharmacological inhibition of MARK4 in a clinical setting provide evidence for its therapeutic potential?
Given the male predominance in the study, how does MARK4 expression differ between men and women?
Could estrogen or androgen signaling modulate MARK4 expression and its impact on heart failure?
Is the observed increase in MARK4 due to chronic intermittent hypoxia, or are other factors in SDB contributing to its upregulation?
Would CPAP therapy or oxygen supplementation reduce MARK4 expression and improve cardiac function?
Reviewer 2 Report
Comments and Suggestions for Authors
In the study by Bettina Seydel et al., titled “Increased Myocardial MARK4 Expression in Patients with Heart Failure and Sleep-disordered Breathing”, the authors investigate the expression of MARK4 in a clinical population. MARK4 is a serine/threonine kinase involved in cytoskeletal regulation, cell cycle control, apoptosis, and metabolic pathways. Of particular interest is its role in the suppression of the AMPK pathway and activation of the mTOR pathway, which may contribute to impaired cardiac metabolism and diastolic dysfunction. Thus, the concept of evaluating MARK4 activity in a clinical setting is important and of potential clinical relevance. However, there are several critical issues that should be addressed to strengthen the validity and interpretation of the findings.
Major Concerns:
1. Study Design and Dichotomization of MARK4 Expression:
The primary limitation of this study lies in its analytical approach. The authors dichotomize the cohort based on the median value of MARK4 expression. While this may simplify statistical comparisons, such a division can be problematic, especially if MARK4 expression does not follow a normal distribution. This approach risks comparing biologically dissimilar groups and may introduce bias. A justification for this method, supported by distributional data, is essential.
2. Multiplicity in Subgroup Analyses:
The authors further stratify the data using ejection fraction (EF) and oxygen saturation (SpOâ‚‚) levels. Conducting multiple subgroup analyses without clear a priori hypotheses increases the risk of multiplicity. To enhance scientific rigor, the study should define specific clinical endpoints and evaluate the association between MARK4 levels and these endpoints, ideally determining meaningful cutoff values through appropriate statistical methods (e.g., ROC analysis).
3. Study Population Characteristics:
The patient cohort appears to be derived from the CONSIDER-AF study (NCT02877745), which prospectively enrolled patients undergoing coronary artery bypass grafting (CABG) to investigate the impact of sleep-disordered breathing on atrial fibrillation and perioperative outcomes. Therefore, all included subjects likely had underlying coronary artery disease. While this is a common comorbidity in heart failure, it nonetheless represents a selected and potentially biased population, limiting the generalizability of the findings to the broader heart failure population. This limitation should be explicitly acknowledged in the Introduction and/or Discussion sections.
4. Inclusion Criteria Regarding Sleep Apnea:
The relationship between sleep apnea and heart failure is well established. Heart failure can predispose patients to central sleep apnea, while obstructive sleep apnea is a known risk factor for heart failure development and progression. Mixed forms of sleep apnea are frequently observed in this patient group. The CONSIDER-AF cohort included patients both with and without sleep apnea. It is therefore essential for the authors to clarify whether the current analysis includes only patients with sleep-disordered breathing or the entire cohort. This information is critical for interpreting the observed associations.
5. Lack of Background on Sleep Apnea in the Introduction:
The Introduction does not provide sufficient context regarding sleep-disordered breathing, which is a central component of this study. Given the title and focus of the analysis, a brief overview of the clinical relevance of sleep apnea in heart failure should be added to the Introduction to better frame the rationale and importance of the research.
Round 2
Reviewer 2 Report
Comments and Suggestions for Authors
The author, Dr. Elisabetta Tonet, should be commended for successfully revising and organizing the manuscript into an improved version. The authors have made significant progress in addressing my previous concerns in this revision; however, a key issue remains unresolved.
The response to Concern #1 is inadequate. The revised manuscript still lacks a clear methodological justification for the selection of the median as a cutoff point, and for example, it does not sufficiently explain why the mean was not used instead. Ideally, the variable in question should be treated as a continuous variable and analyzed using multivariable regression to assess its role as a potential risk factor. While it is not uncommon to encounter dichotomization using the median in studies with limited sample sizes—particularly when the prognostic value of a parameter is uncertain—this approach is methodologically problematic and may lead to biased or misleading conclusions. I strongly recommend that the authors explicitly acknowledge this issue as a limitation of the study.
Author Response
Response: We thank the Reviewer for this helpful comment. We refrained from using the mean as a cutoff because gene expression data (also MARK4 expression levels) often exhibit extreme values. Under such circumstances, the mean can be disproportionately influenced by outliers, potentially leading to biased grouping that does not accurately represent the central tendency of the underlying population. The median, by contrast, represents the central data point unaffected by these extremes. However, using the median may also entail biased conclusions, which is a limitation of the study that we now acknowledge in the revised version of the manuscript. Therefore, we have also incorporated regression analyses that incorporate MARK4 expression as a continuous variable (Figure 2 and Figure 3) and are in accordance with the analyses based on the dichotomization of MARK4. This further strengthens the conclusion that myocardial MARK4 expression is increased in patients with a lower LVEF and in SDB.
We included text on page 13 in lines 348-359, addressing this issue and explicitly acknowledging it as a limitation:
“As this study is a retrospective analysis, it entails several limitations. First, we stratified groups based on the median MARK4 expression. The median was specifically selected as a cutoff because it is more robust against extreme values, which are commonly encountered in biological data such as gene expression levels. While this simplifies statistical comparisons, such an approach may introduce bias, especially when a variable does not follow a normal distribution, which is the case for MARK4 and a limitation of this study. To address this concern, we have incorporated several additional measures. First, we also compared MARK4 expression in patient groups stratified by the LVEF as well as stratified by the presence of SDB. Plus, we performed various uni- and multivariable regression analyses based on the continuous MARK4 expression levels. Notably, all these analyses support our conclusion that MARK4 expression is increased in patients with a lower LVEF and in SDB.”.